# Non-Steroidal Anti-Inflammatory Drugs (NSAIDs): Usage and co-prescription with other potentially interacting drugs in elderly: A cross-sectional study

Nuru Abdu[1]*, Asmerom Mosazghi[1], Samuel Teweldemedhin[1], Luwam Asfaha[1], Makda Teshale[1], Mikal Kibreab[1], Indermeet Singh Anand[1], Eyasu H. Tesfamariam[2], Mulugeta Russom[3]

1 School of Pharmacy, Asmara College of Health Sciences, Asmara, Eritrea, 2 Department of Statistics, Biostatistics and Epidemiology Unit, College of Science, Eritrean Institute of Technology, Mai Nefhi, Eritrea, 3 Eritrean Pharmacovigilance Center, National Medicines and Food Administration, Asmara, Eritrea

* pharmacistnuru@gmail.com

## Abstract

Globally, usage of non-steroidal anti-inflammatory drugs (NSAIDs) in elderly with chronic pain has been reported as frequent. Though NSAIDs are fundamental in maintaining their quality of life, the risk of polypharmacy, drug interactions and adverse effects is of paramount importance as the elderly usually require multiple medications for their co-morbidities. If prescriptions are not appropriately monitored and managed, they are likely to expose patients to serious drug interactions and potentially fatal adverse effects. This study was conducted to assess the appropriateness of NSAIDs use and determine the risk of NSAIDs related potential interactions in elderly. An analytical cross-sectional study was conducted among elderly out-patients (aged 60 and above) who visited three hospitals in Asmara, Eritrea, between August 22 and September 29, 2018. A stratified random sampling design was employed and data was collected using an interview-based questionnaire and by abstracting information from patients' prescriptions and medical cards. Descriptive and analytical statistics including chi-square test and logistic regression were employed using IBM SPSS (version 22). A total of 285 respondents were enrolled in the study with similar male to female ratio. One in four of all respondents were chronic NSAIDs users and NSAIDs risk practice was reported in 24%. Using chronic NSAIDs without prophylactic gastro-protective agents, self-medication, polypharmacy and drug-drug interactions were the main problems identified. A total of 322 potential interactions in 205 patients were identified and of which, 97.2% were classified as moderate, 0.6% severe and the rest were mild. Those who involved in self-medication were more likely to be exposed to drug interactions. Diabetes (AOR = 2.39, 95% CI: 1.14, 5.02) and hypertension (AOR = 9.06, 95% CI: 4.00, 20.51) were associated with chronic NSAIDs use and incidence of potential drug interactions (AOR = 3.5, 95%CI: 1.68, 4.3; AOR = 2.81, 95%CI: 1.61, 4.9 respectively), while diabetes AOR = 4.5, 95% CI: 2.43, 8.35) and cardiac problems (AOR = 4.29, 95% CI: 1.17, 15.73) were more likely to be associated with incidence of polypharmacy. In conclusion, chronic use of NSAIDs without gastro-protective agents and therapeutic duplication of NSAIDs were

**Data Availability Statement:** The complete de-identified data set used for the conclusions of this study is available at: https://osf.io/nm8ue/.

**Funding:** The author(s) received no specific funding for this work.

**Competing interests:** The authors have declared that no competing interests exist.

commonly which requires attention from programmers, health facility managers and healthcare professionals to safeguard elderlies from preventable harm.

## Introduction

Non-steroidal anti-inflammatory drugs (NSAIDs) are used all over the world for their analgesic, anti-inflammatory, and antipyretic effects [1]. NSAIDs are among the most commonly prescribed class of medications globally and they account for approximately 5–10% of all medications prescribed each year [2]. For obvious reasons, elderly are among the frequent users of NSAIDs [3–5] and the fact that these sub-population are highly involved in prescription and non-prescription medications [6], they are highly susceptible to polypharmacy, drug-drug interactions and ultimately drug related complications and even death [7–9]. Serious/fatal gastrointestinal problems including ulcer and bleeding have been frequently reported with chronic use of NSAIDs [10] and thus, co-prescription of gastro-protective agents has paramount importance in preventing such risks [11]. In the elderly, it was estimated that 29% of fatal peptic ulcer complications were possibly due to NSAIDs [12]. Despite this fact, gastro-protective agents were poorly co-prescribed along with NSAIDs [13] and the other serious adverse effects reported with NSAIDs even amplify this concern.

Use of multiple drugs per prescription (polypharmacy) is recognized as independent risk factor for serious adverse drug reactions in the elderly [14, 15]. On the other hand, the clinician's perception of the clinical relevance of drug-drug interactions is not fully appreciated [16, 17]; thus, underestimating the relevant risk when multiple drugs are co-administered. Though polypharmacy might be inevitable in these group of populations, clinicians need to follow recent guidelines and continually update their knowledge on potential interactions, safety signals and their risk mitigation strategies.

In clinical practice, there is an important gap between what is theoretically known and practical exercises in the ground [7]. In Eritrea, to the authors knowledge, there no studies conducted so far to evaluate the appropriateness of the use of NSAIDs in elderlies. Due to shortage of physicians, lower health cadres are authorized to prescribe medicines and recent studies show that self-medication and dispensing non-over-the-counter medicines without prescription is a common practice [18, 19].

All the aforementioned factors contributed to the requirement of further research and stricter control on the use of NSAIDs in elderly. This study is therefore conducted to assess the appropriateness of NSAIDs usage and determine the risk of potential drug interactions with NSAIDs in elderlies in selected hospitals in Asmara, Eritrea.

## Materials and methods

### Study design and setting

An analytical cross-sectional study with a quantitative approach was conducted in three selected hospitals Asmara, the capital, namely: Halibet national referral hospital, Sembel hospital (private) and Bet-Mekae community hospital. Data was collected between August 22 and September 29, 2018 for a period of 30 working days.

### Study and source population

Elderly patients, aged 60 years and above, taking one or more NSAIDs who attended the study sites during the study period formed the study population. Elderly patients, regardless of their sex, who were clinically stable and willing to provide consent to be part of the study were

eligible. Subjects with illegibly written prescriptions, those unwilling to participate or with obvious debilitating conditions and who couldn't pass on reliable information were excluded. The study has no specific source population as one of the selected hospitals was a national referral hospital which follows patients referred or self-referred from other regions.

## Sampling design

In order to get representative samples from each hospital, stratified random sampling was utilized. The three hospitals were considered as strata, and participants were selected using systematic random sampling because of the unavailability of prior information on patient visits.

## Sample size determination

Sample size was computed by considering the finite population correction factor: $n = NZ^2pq/[pqZ^2 + (N-1)d^2]$. The total sample size (n) was calculated using the following assumptions: expected proportion of elderly patients with drug interaction (p) and those without drug interaction (q) were taken as 0.5, Z statistic for 95% level of confidence (Z = 1.96), estimated population size (N) of 900, margin of error (d) of 0.05 and 10% non-response rate. Considering the above assumptions, the final sample size was found to be 297.

## Data collection tools

A data collection form (S1 File) comprising of five sections was used. The data collection form was self-developed and further reviewed using panel of experts in the fields of pharmacy, pharmacoepidemiology and medicine. The interviewers were fifth-year pharmacy students trained in a one-day workshop to ensure perspicuity of the items so as to maximize the within and between inter-rater consistencies. Section A, includes socio-demographic and background characteristics of the patients' such as age, sex, marital status, educational level, religion, ethnic group, chronic illness and history of gastrointestinal upset. Section B, encompasses, five questions that assess usage of gastro-protective agents among the chronic NSAID users and adverse drug reactions encountered. Section C intends to record information of the prescribed NSAIDs from patients' prescriptions. This information includes dose, frequency, duration of treatment, route of administration and dosage form. Section D was aimed at recording name of the prescribed and self-medicated drugs for analysis of drug-drug interactions and section E was used to record information from patients' medical cards like indication(s) of the prescribed NSAID(s), disease status, history of co-morbidities and history of gastrointestinal upset of the patients'. Potential drug interaction was evaluated using drugs.com [20] and WebMD [21] on October 2018.

## Data collection procedure

The investigators explained purposes of the study to the participants and those who gave consent were enrolled. Exit interview was conducted for each patient using a questionnaire. The exit interview was aimed at exploring information on patients' socio-demographic and background characteristics, co-prescription of gastro-protective agents with chronic NSAIDs use, adverse effects encountered, medical history and self-medication status. Then, information contained in their prescriptions were recorded and their medical cards were assessed to document their co-morbid conditions, indication(s) of the prescribed NSAIDs and history of GI upset. Finally, the potential drug-drug interactions were screened using www.drugs.com drug interaction checker. WebMD drug interaction checker was used if information on potential interaction is unavailable in drugs.com interaction checker [Fig 1]. All the obtained data were documented and no follow up was made due to the cross-sectional nature of the study.

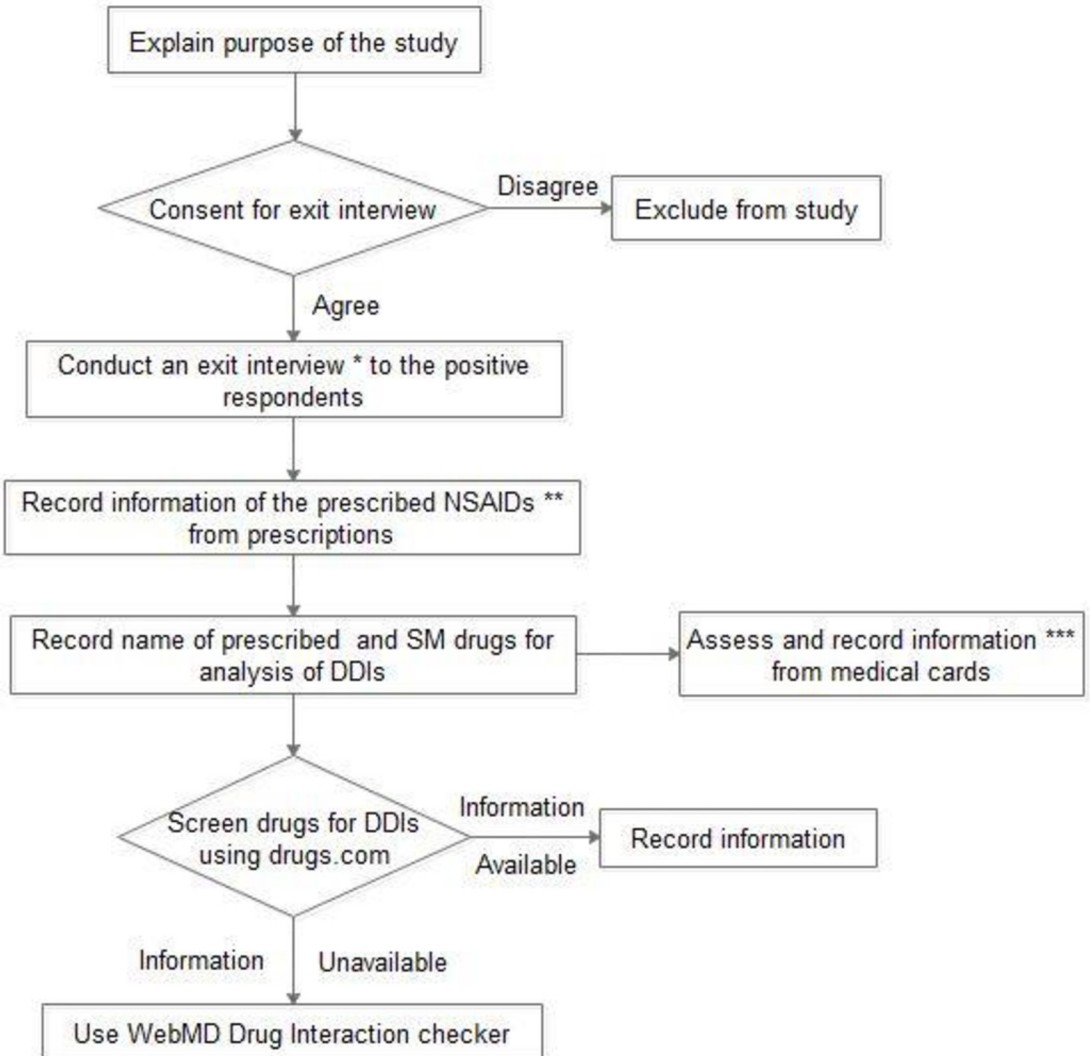

**Fig 1. Data collection procedure.** Exit interview * aimed at exploring socio-demographic and background characteristics, usage of gastro-protective agents among chronic NSAID users, ADRs encountered and self-medication status of the participants. Information of the prescribed NSAIDs ** includes dose, frequency, duration of treatment, route of administration and dosage form. Information from medical cards *** includes indications of the prescribed NSAID(s), comorbidity and history of peptic ulcer. *NSAIDs*: Non-steroidal Anti-Inflammatory Drugs; *SM* drugs: Self-Medicated drugs; *DDIs*: Drug-drug interactions.

## Pre-test

A pre-test was conducted on 31 participants from 17 to 21 August, 2018, to ensure comprehensibility, compatibility of the questionnaire and to familiarize data collectors at two randomly selected hospitals. Prior to the pre-test, a one-day orientation workshop was provided to the data collectors and supervisors. Based on the experience gathered in the pre-test, the questionnaire was revised and used for the actual data collection.

## Ethical consideration

Ethical approval was obtained from the Ministry of Health research ethics and protocol review committee and Asmara College of Health Sciences ethical clearance committee. Besides, permission was obtained beforehand from the medical directors and head of pharmacies of the

respective hospitals. Study participants were informed about the objective of the study and written informed consent was obtained from each respondent. During data entry and analysis, patient identifiers were anonymized and all the information gathered was kept in strict confidence and used only for this study's purpose.

## Statistical analysis

The collected data were double entered on the Census and Survey Processing system-7.0 (CSPro-7.0) to minimize keying errors and was exported to Statistical Package for Social Science-22 (SPSS-22) for statistical analysis. Descriptive summaries of the socio-demographic variables were computed using mean (with standard deviation) or median (with interquartile range) as appropriate. Associates of polypharmacy, drug interact were discovered using logistic regression. Furthermore, factors that were related to chronic NSAIDs use were identified using bivariate logistic regression. Chi-square test and logistic regression were used to explore existence of trend and magnitude of possible associations. Odds ratio with 95% confidence interval was reported in all logistic regression analyses. All analyses were considered significant when $p < 0.05$.

## Operational definitions

**Mild-interaction.** An interaction is considered 'mild' if it has minimal clinical significance. Manifestations may include an increase in frequency or severity of the side effects but generally would not require a major alteration in therapy [20, 21].

**Moderate-interaction.** An interaction is considered 'moderate' if it has clinical significance of moderate importance. The interaction may result in exacerbations of the patient's condition and/or require an alteration in therapy [20, 21].

**Severe-interaction.** An interaction is considered 'severe' if it is highly clinically significant. The interaction may be life-threatening and/or require medical intervention to minimize or prevent serious adverse effects [20, 21].

**Chronic NSAID users.** A patient can be considered a chronic NSAID user if he/she consumes NSAIDs for at least three months [22].

**Polypharmacy.** Is defined as the concomitant use of four or more drugs prescribed at the same time [23].

**Prescription pattern.** Includes information regarding type of NSAIDs, dosage form, route of administration of NSAIDs and number of drugs ordered per prescription.

**Risky practice.** A respondent and/or prescriber was considered at risky practice if respondents self-medicated themselves along with potentially interacting prescribed drugs, or if prescribers fail to prescribe gastro-protective agents for those with previous history of gastrointestinal upset and/or chronic use NSAIDs. Prescribers ordering two or more NSAIDs at the same time were also considered at risky practice.

## Results

### Socio-demographic characteristics and background characteristics

Data collectors were able to approach 297 subjects in the three hospitals during the study period. However, 12 subjects were excluded from the study for different reasons and a total of 285 respondents with a median age of 69 years (IQR: 15) were enrolled in the study [Fig 2].

About three-fourth (78.2%) of the respondents had not completed high school. Majority of the respondents (66.7%) had chronic illnesses and the most common chronic illnesses reported were hypertension (52.3%) and diabetes (29.5%). About 35% of all the respondents

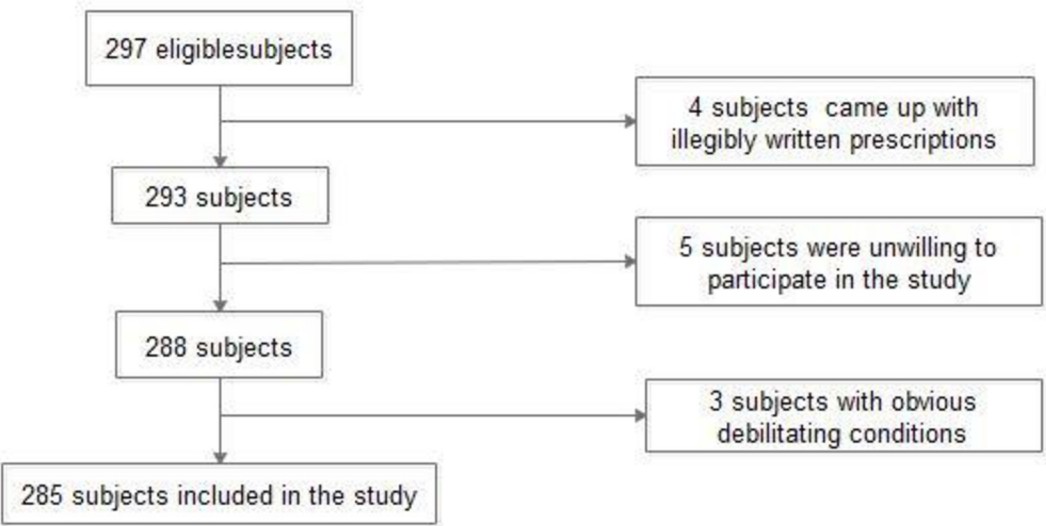

**Fig 2. Study participants that were eligible and finally included in the study.**

had history of gastro-intestinal upset [Table 1]. A detailed socio-demographic characteristics of the study population is depicted in Table 1.

The most common reasons for prescription of NSAIDs were: anti-platelet effect (36.2%) with low dose aspirin, arthritis (5.35%), backache (4.95%), knee pain (3%), and leg pain (2.3%). The mean number of drugs per prescription was 2.61 and 23.9% of the prescriptions had two or more NSAIDs per prescription [Table 2]. The most prescribed NSAIDs were aspirin (36.5%) and diclofenac (36.5%) followed by indomethacin (22.5%) and Ibuprofen (22.5%).

Of those with previous history of gastrointestinal upset (101/285), concomitant use with gastro-protective agents was documented in 9.9%, which was mainly omeprazole (8.9%).

### Incidence of polypharmacy among elderly NSAIDs users and associated risk factors

Out of the 285 respondents, 20% were exposed to polypharmacy. Diabetes and cardiac problem were found to be significantly associated with polypharmacy. Patients with diabetes (AOR = 4.5, 95% CI: 2.43, 8.35) and cardiac problems (AOR = 4.29, 95% CI: 1.17, 15.73) were more likely to be exposed to poly-pharmacy [Table 3].

### Usage of gastro-protective agents among chronic NSAIDs users

Majority (59.3%) of the respondents had history of NSAIDs use, of whom 42% (71/169) were chronic users of NSAIDs. Gastro-protective agents were co-prescribed in 25.4% (18/71) of those with chronic use of NSAIDs. Omeprazole (n = 11), antacid (n = 6) and famotidine (n = 1) were prescribed as gastro-protective agents. Self-reported adverse drug reactions were documented in 12 (16.9%) of the patients on chronic use of NSAIDs that were dominated by gastrointestinal upset.

Respondents who had either diabetes (AOR = 2.39, 95% CI: 1.14, 5.02) or hypertension (AOR = 9.06, 95% CI: 4.00, 20.51) were more likely to be chronic users of NSAIDs [Table 4].

### Analysis of NSAIDs related drug interactions and its associated risk factors

The number of respondents who reportedly self-medicated themselves were 26.7%. Potential NSAIDs related interactions with self-medication was observed in 37 respondents, giving a

**Table 1. Socio-demographic and background characteristics of the respondents (N = 285).**

| Variable | Category | Number | Percent |
|---|---|---|---|
| Age (Md; IQR = 69.00,15) | | | |
| | 60 to 69 | 144 | 50.5 |
| | 70 to 79 | 91 | 32 |
| | 80 or above | 50 | 17.5 |
| Sex | Male | 143 | 50.2 |
| | Female | 142 | 49.8 |
| Marital Status | | | |
| | Married | 171 | 60 |
| | Single | 9 | 3.2 |
| | Divorced | 6 | 2.1 |
| | Widowed | 99 | 34.7 |
| Level of Education | | | |
| | Illiterate | 97 | 34 |
| | Primary | 73 | 25.6 |
| | Middle | 53 | 18.6 |
| | Secondary or higher | 62 | 21.7 |
| Occupation | | | |
| | Governmental | 49 | 17.2 |
| | Private | 26 | 9.1 |
| | Self-employed | 40 | 14 |
| | Unemployed | 54 | 18.9 |
| | House wife | 116 | 40.7 |
| Chronic illness[‡] | Hypertension | 149 | 52.3 |
| | Diabetes | 84 | 29.5 |
| | Asthma | 14 | 4.9 |
| | Renal Failure | 2 | 0.7 |
| | Cardiac problem | 11 | 3.9 |
| | Others[*] | 4 | 1.4 |
| | No chronic illness | 95 | 33.3 |
| History of GI upset | | | |
| | Yes | 101 | 35.4 |
| | No | 184 | 64.6 |

Md: Median; IQR: Interquartile range.

[‡]Total percent might exceed 100 due to multiple answers.

[*]Others include cancer and rheumatoid arthritis.

total of 41 potential drug interactions varying in severity [Fig 3]. Of all who were self-medicated themselves, 48.7% were at risky practice as they were using self-medicated with other drugs that have potential interactions with NSAIDs.

On the other hand, 67.1% (n = 51/76) of the self-medicated drugs were NSAIDs which had potential interaction with other prescribed drugs apart from the prescribed NSAIDs. Potential interactions were observed in 19 respondents (n = 19/51), giving a total of 24 potential drug interactions. All were classified as moderate.

Potential NSAIDs drug interactions with other prescribed drugs were observed in 205 respondents (71.9%), giving a total of 322 potential drug interactions, of which 0.6% were classified as severe, 97.2% moderate and 2.2% mild.

**Table 2. Pattern of NSAIDs prescription among the elderly (N = 285).**

| Variable | | Number | Percent |
|---|---|---|---|
| Number of NSAIDs per prescription | | | |
| | 1 | 217 | 76.1 |
| | 2 | 60 | 21.1 |
| | 3 | 8 | 2.8 |
| Total number of drugs per prescription (Md = 2; IQR = 1) | | | |
| | 1 | 31 | 10.9 |
| | 2 | 122 | 42.8 |
| | 3 | 75 | 26.3 |
| | ≥4 | 57 | 20 |

Md: Median; IQR: Interquartile range.

The NSAIDs with the greatest risk of drug interactions were aspirin (n = 168), indomethacin (n = 52), ibuprofen (n = 46) and diclofenac tablet (n = 41). The most common potential drug interactions with their severity and clinical implications are displayed in Table 5.

Chi-square test for trend analysis indicated there was significant increase in interaction with increase the number of drugs prescribed ($\chi2$ = 20.72, P<0.001). As the number of drugs prescribed increases by one unit, the odds of interactions increase by 3.25 unit (COR: 3.25; 95%CI: 1.89, 5.61) [Table 6].

**Table 3. Association of polypharmacy with age, gender and chronic illness.**

| Variables | | Bivariate analysis | | Multivariate analysis | |
|---|---|---|---|---|---|
| | | Crude OR (95% CI) | *p*-value | Adjusted OR (95% CI) | *p*-value |
| Age | | | | | |
| | 60 to 69 | *Ref.* | 0.511 | - | - |
| | 70 to 79 | 1.34 (0.70, 2.59) | 0.382 | - | - |
| | 80 or above | 1.50 (0.69, 3.78) | 0.305 | - | - |
| Sex | | | | | |
| | Male | 1.13 (0.63, 2.02) | 0.678 | - | - |
| | Female | *Ref.* | | - | - |
| Hypertension | | | | | |
| | Yes | 1.75 (0.96, 3.17) | 0.068 | - | - |
| | No | *Ref.* | | - | - |
| Diabetes | | | | | |
| | Yes | 4.33 (2.36, 7.96) | <0.001 | 4.5 (2.43, 8.35) | <0.001 |
| | No | *Ref.* | | *Ref.* | |
| Asthma | | | | | |
| | Yes | 1.53 (0.33,7.03) | 0.586 | - | - |
| | No | *Ref.* | | - | - |
| Renal Failure | | | | | |
| | Yes | 4.05 (0.25,65.81) | 0.325 | - | - |
| | No | *Ref.* | | - | - |
| Cardiac Problems | | | | | |
| | Yes | 3.56 (1.05,12.11) | 0.042 | 4.29 (1.17, 15.73) | 0.028 |
| | No | *Ref.* | | *Ref.* | |

OR: Odds Ratio, CI: Confidence Interval, Ref: Reference.

**Table 4. Associations of chronic NSAID users with age, gender, and chronic illnesses.**

| Variables | | Bivariate analysis | | Multivariate analysis | |
|---|---|---|---|---|---|
| | | COR (95% CI) | *p*-value | AOR (95% CI) | *p*-value |
| Age | | | | | |
| | 60 to 69 | *Ref.* | 0.592 | - | - |
| | 70 to 79 | 1.21 (0.59, 2.46) | 0.6 | - | - |
| | 80 or above | 1.51 (0.68, 3.34) | 0.314 | - | - |
| Sex | | | | | |
| | Male | 1.01 (0.55, 1.87) | 0.968 | - | - |
| | Female | *Ref.* | | - | - |
| Hypertension | | | | | |
| | Yes | 9.99 (4.46, 22.38) | <0.001 | 9.06 (4, 20.51) | <0.001 |
| | No | *Ref.* | | *Ref.* | |
| Diabetes | | | | | |
| | Yes | 3 (1.54, 5.84) | 0.001 | 2.39 (1.14, 5.02) | 0.022 |
| | No | *Ref.* | | *Ref.* | |
| Asthma | | | | | |
| | Yes | 0.68 (0.16, 2.8) | 0.59 | - | - |
| | No | *Ref.* | | - | - |
| Renal Failure | | | | | - |
| | Yes | 0 (0) | 0.99 | - | - |
| | No | *Ref.* | | - | - |
| Cardiac Problem | | | | | |
| | Yes | 0.54 (0.10, 2.86) | 0.468 | - | - |
| | No | *Ref.* | | - | - |

OR: Odds Ratio, CI: Confidence Interval, Ref: Reference.

Respondents which had either diabetes (AOR = 3.5, 95%CI: 1.68, 4.3) or hypertension (AOR = 2.81, 95%CI: 1.61, 4.9) were found to be significantly associated with NSAIDs potential drug interactions [Table 7].

## Discussion

In this study, one in four of the respondents had two or more NSAIDs per prescription. Even though this is much lower than that reported by Jayakumari et al. (77.3%) [24], its implication on the consumers could be devastating duet to potentially serious risk of drug-drug interactions and adverse drug reactions with no additional therapeutic value. Though a substantial number of respondents had history of gastrointestinal upset and were on chronic use of

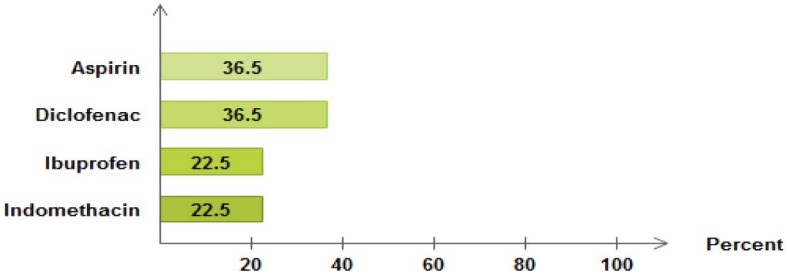

**Fig 3. Severity of NSAIDs potential interactions with self-medicated drugs.**

**Table 5. Most common potential drug interactions with their severity and clinical implications.**

| Drug interactions | | Severity | Clinical implication | Frequency (n) |
|---|---|---|---|---|
| Drug 1 (NSAID) | Drug 2 | | | |
| Aspirin | Ibuprofen | Severe | Antagonization of anti-platelet and cardio-protective effect | 2 |
| Aspirin | Enalapril | Moderate | Attenuation of hypotensive and vasodilator effect | 66 |
| Aspirin | Glimepiride | Moderate | Increased risk of hypoglycemia | 24 |
| Aspirin | *Hydrochlorothiazide* | Moderate | Increased anti-platelet effect | 23 |
| Aspirin | Insulin | Moderate | Increased risk of hypoglycemia | 15 |
| Indomethacin | Methylprednisolone | Moderate | Risk of gastrointestinal toxicity | 15 |

NSAID therapeutic duplication due to prescribed and self-medicated drugs was also detected in 28.8% of the respondents.

NSAIDs, use of gastro-protective agents was found to be very poor which is against the international guidelines and recommendations [11, 25, 26]. Gastro-protective agents were prescribed in only a quarter of the chronic NSAID users. This finding was higher than the finding of a similar study in UK (10%) [13] but much lower than that reported in US (99.8%) [27]. The possible explanation for the variation in results maybe the various prescription habits among countries and the level of knowledge about the concurrent use and importance of gastro-protective agents in preventing or minimizing NSAIDs-induced gastro-intestinal complications.

Potential drug-drug interactions of NSAIDs with other prescribed drugs was also found to be significant. Polypharmacy and self-medication were identified as the main determinants of the drug interaction. Some of those who were more involved in self-medication were prone to potentially severe drug interactions and majority were exposed to interactions having moderate clinical significance.

The implication is that, therapeutic duplication of NSAIDs and polypharmacy would expose elderlies to serious or potentially fatal adverse effects including nephrotic syndrome [28], acute renal failure [9], heart failure [29] and gastrointestinal problems [10]. Furthermore, concurrent use of some NSAIDs like ibuprofen, can interfere with the antiplatelet effect of low dose aspirin by blocking aspirin's irreversible cyclooxygenase-1 inhibition [29]. NSAIDs-related complications could also compromise adherence of other therapeutic agents used for chronic diseases.

Taking the age of the study population into consideration, polypharmacy might be inevitable in many patients. Prescribers should, however, responsibly take medication history, avoid prescriptions of unnecessary medicines and pharmacists need to counsel elderlies to refrain from self-medication. When at times polypharmacy becomes inevitable, a close and intensive monitoring, using multidisciplinary approach, is required to prevent serious drug-drug interactions, drug-disease interactions and adverse effects. Immediate attention from program managers and policy makers are also required to introduce risk mitigation strategies that could protect patients from preventable harm.

## Limitation of the study

Due to the cross-sectional nature of the study, all drug-drug interactions documented in this study are theoretical and thus, their clinical significance at ground might be over-or under-

**Table 6. Association between number of drugs prescribed and potential drug interactions.**

| Description | Total number of drugs in prescription | | | | | Linear-by-Linear Association | |
|---|---|---|---|---|---|---|---|
| | 2 | 3 | 4 | 5 | 7 | χ2-value | *p*-value |
| Occurrence of interaction (%) | 67.2 | 89.6 | 93 | 100 | 100 | 20.72 | <0.001 |

**Table 7. Associations of NSAID-related drug interactions with age, gender and clinical conditions.**

| Variable | | Bivariate analysis | | Multivariate analysis | |
|---|---|---|---|---|---|
| | | Crude OR (95% CI) | *P*-value | Adjusted OR (95% CI) | *P*-value |
| Age | | | | | |
| | 60 to 69 | *Ref.* | | - | - |
| | 70 to 79 | 1.34 (0.75, 2.42) | 0.326 | - | - |
| | 80 or above | 1.44 (0.69, 3.01) | 0.334 | - | - |
| Sex | | | | | |
| | Male | 0.88 (0.53, 1.47) | 0.624 | - | - |
| | Female | *Ref.* | | - | - |
| Hypertension | | | | | |
| | Yes | 3.12 (1.81, 5.33) | <0.001 | 2.81 (1.61, 4.9) | <0.001 |
| | No | *Ref.* | | *Ref.* | |
| Diabetes | | | | | |
| | Yes | 3.95 (1.92, 8.13) | <0.001 | 3.5 (1.68, 4.3) | 0.001 |
| | No | Ref. | | *Ref.* | |
| Asthma | | | | | |
| | Yes | 0.97 (0.3, 3.2) | 0.966 | - | - |
| | No | *Ref.* | | - | - |
| Renal Failure | | | | | |
| | Yes | - [‡] | 0.99 | - | - |
| | No | *Ref.* | | - | - |
| Cardiac Problems | | | | | |
| | Yes | 1.79 (0.38, 8.48) | 0.463 | - | - |
| | No | *Ref.* | | - | - |

OR: Odds Ratio, CI: Confidence Interval, Ref: Reference.

[‡]One out of four cells has zero observed count.

estimated. In addition, the adverse effects and history of self-medication presented in this study were all self-reported which might be subjected to recall bias. Incompleteness of information in medical cards, and NSAIDs supply inconsistencies due to stock-outs were some of the limitations of the study. The small sample size might also limit the statistical power of the analysis performed.

## Conclusion and recommendations

Chronic use of NSAIDs without prophylactic gastro-protective agents, therapeutic duplication of NSAIDs and polypharmacy were the major problems in this study. To minimize complications, where possible, the lowest effective dose of NSAIDs should be prescribed for the shortest possible time. Besides, regular updating of national standard treatment guidelines and formularies, use of gastro-protective agents for chronic NSAID users, introduction of electronic medical records for tracing drug interactions and awareness raising programs are highly recommended.

## Supporting information

**S1 File.**
(PDF)

## Acknowledgments

We would like to forward our sincere gratitude to Dr. Araia Brahane, Dr. Saud Mohammed, Dr. Yosief Yohannes, N. Saleem Basha, Bruk Woldai and Dawit Tesfai who were involved in the face and content validation of our questionnaire. We also sincerely thank Dr. Luul Banteyrga (Medical Director of Halibet Hospital), Dr. Yosief Yacob (Medical Director of Sembel Hospital) and Dr. Tsegereda Mehari (Medical Director of Bet-Mekae Community Hospital) who have warmly accepted and approved the study to be conducted in their hospitals. Finally, we would also like to thank all participants of this study for being cooperative in the process.

## Author Contributions

**Conceptualization:** Mulugeta Russom.

**Data curation:** Nuru Abdu, Asmerom Mosazghi, Luwam Asfaha, Eyasu H. Tesfamariam.

**Formal analysis:** Nuru Abdu, Asmerom Mosazghi, Samuel Teweldemedhin, Luwam Asfaha, Makda Teshale, Mikal Kibreab, Indermeet Singh Anand, Eyasu H. Tesfamariam, Mulugeta Russom.

**Methodology:** Nuru Abdu, Asmerom Mosazghi, Samuel Teweldemedhin, Luwam Asfaha, Makda Teshale, Mikal Kibreab, Indermeet Singh Anand, Eyasu H. Tesfamariam, Mulugeta Russom.

**Project administration:** Mulugeta Russom.

**Supervision:** Indermeet Singh Anand, Eyasu H. Tesfamariam, Mulugeta Russom.

**Writing – original draft:** Nuru Abdu, Asmerom Mosazghi, Samuel Teweldemedhin, Luwam Asfaha, Makda Teshale, Mikal Kibreab.

**Writing – review & editing:** Nuru Abdu, Asmerom Mosazghi, Samuel Teweldemedhin, Luwam Asfaha, Makda Teshale, Mikal Kibreab, Indermeet Singh Anand, Eyasu H. Tesfamariam, Mulugeta Russom.

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
