## [Decision Letter · Decision Letter 0]

20 Apr 2020

PONE-D-20-05326

Non-Steroidal Anti-Inflammatory Drugs (NSAIDs): Usage and Co-Prescription with Other Potentially Interacting Drugs in Elderly: a cross-sectional study

PLOS ONE

Dear mr Abdu,

Thank you for submitting your manuscript to PLOS ONE. After careful consideration, we feel that it has merit but does not fully meet PLOS ONE’s publication criteria as it currently stands. Therefore, we invite you to submit a revised version of the manuscript that addresses the points raised during the review process.

We would appreciate receiving your revised manuscript by Jun 04 2020 11:59PM. To enhance the reproducibility of your results, we recommend that if applicable you deposit your laboratory protocols in protocols.io, where a protocol can be assigned its own identifier (DOI) such that it can be cited independently in the future. For instructions see: http://journals.plos.org/plosone/s/submission-guidelines#loc-laboratory-protocols

We look forward to receiving your revised manuscript.

Kind regards,

Jinn-Moon Yang

Academic Editor

PLOS ONE

3. Please ensure that you refer to Figure 4-5 in your text as, if accepted, production will need this reference to link the reader to the figure.

4. We note you have included a table to which you do not refer in the text of your manuscript. Please ensure that you refer to Table 6 in your text; if accepted, production will need this reference to link the reader to the Table.

Reviewers' comments:

Reviewer's Responses to Questions

**Comments to the Author**

1. Is the manuscript technically sound, and do the data support the conclusions?

Reviewer #1: Partly

Reviewer #2: Partly

2. Has the statistical analysis been performed appropriately and rigorously? 

Reviewer #1: No

Reviewer #2: N/A

3. Have the authors made all data underlying the findings in their manuscript fully available?

Reviewer #1: Yes

Reviewer #2: No

4. Is the manuscript presented in an intelligible fashion and written in standard English?

Reviewer #1: No

Reviewer #2: No

5. Review Comments to the Author

Reviewer #1: The authors proposed a cross sectional study in elderly population to describe prevalence of NSAIDs prescription, and potential drug-drug interactions. In this study, the authors also analyzed associations between subject conditions and polypharmacy/chronic NSAIDs usage/NSAIDs-related drug interaction. I have several concerns before the manuscript can be considered for publication.

Major issues:

1.Introduction section is too short. Please describe more in subsections including background, literature review, challenges of previous studies and specific aims of this study.

2.Previous studies reviewed in the introduction are not up-to-date. Please discuss more related studies published at least within the past two years.

3.Coherence and readability of this manuscript can be improved furthermore. English editing may be required.

4.What did the authors mean about ‘illegibly written prescriptions’, ‘has no specific source population’, and ‘prescription pattern of NSAIDs during the study period was almost satisfactory’?

5.What were the results of validity and reliability tests of the final questionnaire? Were the interviewers trained to acquire the information? Were they blinded to information from the medical card, particularly the prescribed drugs?

6.Based on the sample size estimation described by the authors, this study would require 297 participants. But, the sample size was 285 participants. Why did the authors not pursue the minimum sample size after attrition? In addition, why did the authors not apply ad hoc power analysis to estimate the sample size?

7.What were the variables adjusted for the multivariable analyses? Sample sizes of the event for polypharmacy, chronic NSAIDs usage and potential drug-drug interaction were 57, 71 and 37 subjects, respectively. Assuming there were 6 other variables to adjust each variable of interest, the samples might lack of power to identify effect of the variable on either polypharmacy or potential drug-drug interaction. In addition, multivariable modelling needs 20 instead of 10 events per variable.

8.What are the authors arguments applying a cross-sectional design for association tests?

 

Minor issues:

Typos, over-/under-reported information, unusual abbreviation, etc. were found in the manuscript. Please correct those. For example:

Line 83 to 85: Why was this information needed?

Line 105-106: Drug database may be frequently changed. When was the database used?

Line 180 and Fig. 2: The numbers of eligible subjects were not matched.

Several lines: Which one was commonly used, AOR or aOR?

Reviewer #2: In this manuscript, the authors aimed to assess the appropriateness of NSAIDs use and incidence of NSAIDs related potential interactions in elderly in Eritrea. The study is interesting, however, a number of issues should be addressed before its publication.

Major issues:

1. In Table 2, the authors listed background information of the respondents (N = 285), including chronic illness and history of GI (gastro-intestinal) upset. More background information of the respondents (N = 285) should be listed in this table, for example, GPA (gastro-protective agents), chronic NSAIDs, SM (self-medicated), risky practice. (Polypharmacy information was listed in Table 3)

2. Since elderly patients (aged 60 years and above) taking one or more NSAIDs who attended the study sites during the study period formed the study population, could the authors explain what the majority represent in the sentence of "Majority (59.3% =? 169/285) of the elderly had history of NSAIDs use and 42% (71/169) of whom were chronic NSAIDs users"?

3. The illustration of risky practice in Figure 4 is not clear enough.

4. In Table 7, the authors did chi-square trend test to evaluate the association between number of drugs prescribed and drug interactions. If the total number of drugs in prescription is one, there will be no interaction for sure. Is it reasonable to include this category in the trend analysis? Please explain the details about how to estimate the odds of interactions increase per unit.

5. The explanatory of analysis of NSAIDs related drug interactions and its associated risk factors was doubtful. Details were not clearly stated.

Minor issues:

There were some errors, for example,

line 180: # of subjects in the three hospitals,

lines 201, 222, 250: # of Figures or Tables,

numbers in Table 7, ... and so on.

6. PLOS authors have the option to publish the peer review history of their article (what does this mean?). If published, this will include your full peer review and any attached files.

Reviewer #1: No

Reviewer #2: No

---

## [Author Response · Author response to Decision Letter 0]

21 Jul 2020

Response to reviewers 

We would like to thank our reviewers for their invaluable inputs and constructive comments that are helpful to massively improve the quality of the manuscript. After careful consideration of the points raised by reviewers, the point-by-point response are as follows: 

Response to Reviewer 1

Major issues

1. Introduction section is too short. Please describe more in subsections including background, literature review, challenges of previous studies and specific aims of this study.

- It is well noted. Efforts are made to incorporate the above comments on the introduction part. Recently published articles related to the subject, gaps in knowledge, and problem statements are incorporated in the introduction part. Please refer changes made on the manuscript. Taking the length of introduction on the impact of readership, we still preferred to limit it to about one page. 

2. Previous studies reviewed in the introduction are not up-to-date. Please discuss more related studies published at least within the past two years.

- This is a very important comment. Our paper has been prepared before one year. Hence, your comment is valuable and we included the recent available studies in the introduction section. 

3. Coherence and readability of this manuscript can be improved furthermore. English editing may be required.

- Accepted. Massive editing is made on the manuscript. We have made 260 insertions, 231 deletions, 10 moves and 153 formatting in the revised manuscript. Accordingly, the reference section is fully revised. Please refer the ‘revised manuscript with track changes’. 

4. What did the authors mean about ‘illegibly written prescriptions’, ‘has no specific source population’, and ‘prescription pattern of NSAIDs during the study period was almost satisfactory’?

- ‘Illegibly written prescriptions’ means the prescriptions which are totally unreadable. 

- ‘Has no specific source population’ refers that one of our hospitals is a national referral and thus we had secondary source population. Meaning, information gathered from that site might not fully represent to the Asmara community. To avoid confusion, we have made some changes in the manuscript. 

- ‘Prescription pattern of NSAIDs during the study period was almost satisfactory’ reflects a subjective judgment and thus, comment is well-taken and revised. Please see changes made in the abstract and conclusion of the main text. 

5. What were the results of validity and reliability tests of the final questionnaire? Were the interviewers trained to acquire the information? Were they blinded to information from the medical card, particularly the prescribed drugs?

A. Validity 

- Face and content validity was performed by involving a panel of experts as stated on the manuscript and accordingly, some changes were made on the questionnaire before it is subjected to pre-test. 

- As to the training of interviewers, it is well noted and added some notes on the manuscript on the orientation workshop. 

B. Reliability of the questionnaire

- As the interviewers were fifth-year pharmacy students and they were trained in a one-day workshop to ensure perspicuity of the items so as to maximize the within and between inter-rater consistencies. This note is included in the manuscript. 

C. Were they blinded to information from the medical card, particularly the prescribed drugs?

- Data collectors were not blinded and had full access to the medical cards. This is explained in the data collection approach section of the manuscript. 

6. Based on the sample size estimation described by the authors, this study would require 297 participants. But, the sample size was 285 participants. Why did the authors not pursue the minimum sample size after attrition? In addition, why did the authors not apply ad hoc power analysis to estimate the sample size?

- In calculation of the sample size, we used a non-response rate of 10% which is approximately 30. Hence, a sample size of 297 less 30 is still tolerable though it is much lower than that. 

7. What were the variables adjusted for the multivariable analyses? Sample sizes of the event for polypharmacy, chronic NSAIDs usage and potential drug-drug interaction were 57, 71 and 37 subjects, respectively. Assuming there were 6 other variables to adjust each variable of interest, the samples might lack of power to identify effect of the variable on either polypharmacy or potential drug-drug interaction. In addition, multivariable modelling needs 20 instead of 10 events per variable.

- Multivariate modelling was done by controlling or adjusting confounding effect upon each variables of interest. Even though we had enough samples on the analysis for interactions of prescribed NSAID with other prescribed drugs (n=205), in some instances the samples size was small that would limit the power of the study. We have reflected in our revision on the limitation section. 

8. What are the authors arguments applying a cross-sectional design for association tests?

- For practicality reasons, we preferred to use cross-sectional study. To our knowledge, cross-sectional study design has limitations on establishing causation but not with associations as association does not imply causation. 

Minor issues

Line 83 to 85: Why was this information needed?

- Accepted. We omitted this information. 

Line 105-106: Drug database may be frequently changed. When was the database used?

- Accepted and date included in the revised manuscript. 

Line 180 and Fig. 2: The numbers of eligible subjects were not matched.

- Accepted. It was type error and the 295 is changed by 297 in the manuscript. 

Several lines: Which one was commonly used, AOR or aOR?

- All reported odds ratio was adjusted (AOR). 

Reviewer 2 

Major issues

1. In Table 2, the authors listed background information of the respondents (N = 285), including chronic illness and history of GI (gastro-intestinal) upset. More background information of the respondents (N = 285) should be listed in this table, for example, GPA (gastro-protective agents), chronic NSAIDs, SM (self-medicated), risky practice. (Polypharmacy information was listed in Table 3)

- It is well noted. Chronic NSAIDs, SM, risky practice and polypharmacy are separate sections of the result part. Instead of two tables, we decided to merge table 1 and 2 and named as ‘socio-demographic and background characteristics of the participants’. 

2. Since elderly patients (aged 60 years and above) taking one or more NSAIDs who attended the study sites during the study period formed the study population, could the authors explain what the majority represent in the sentence of "Majority (59.3% =? 169/285) of the elderly had history of NSAIDs use and 42% (71/169) of whom were chronic NSAIDs users"?

- Accepted. Majority (59.3%) indicate that it’s from the total subjects (285) included in the study.

3. The illustration of risky practice in Figure 4 is not clear enough.

- Accepted. We removed Figure 4 and five from the manuscript as the results are fully reported in the text. 

4. In Table 7, the authors did chi-square trend test to evaluate the association between number of drugs prescribed and drug interactions. If the total number of drugs in prescription is one, there will be no interaction for sure. Is it reasonable to include this category in the trend analysis? Please explain the details about how to estimate the odds of interactions increase per unit.

- Accepted. We excluded prescriptions containing one drug for the trend analysis and we reanalyzed it. Hence, the odds of interaction was changed to 3.25 units, i.e. after excluding prescriptions with only one drug. To estimate the odds of interactions increase per unit, we used bivariate logistic regression. 

5. The explanatory of analysis of NSAIDs related drug interactions and its associated risk factors was doubtful. Details were not clearly stated.

- Accepted. You can see the changes in revised manuscript with track changes. 

Minor changes

- To avoid raised concern by the reviewers, we made serious edit/proof read on the manuscript. Kindly, visit the manuscript. 

Kind regards, 

Nuru Abdu, on behalf of the authors

---

## [Decision Letter · Decision Letter 1]

26 Aug 2020

Non-Steroidal Anti-Inflammatory Drugs (NSAIDs): Usage and Co-Prescription with Other Potentially Interacting Drugs in Elderly: a cross-sectional study

PONE-D-20-05326R1

Dear Dr. Abdu,

We’re pleased to inform you that your manuscript has been judged scientifically suitable for publication and will be formally accepted for publication once it meets all outstanding technical requirements.

Kind regards,

Jinn-Moon Yang

Academic Editor

PLOS ONE

Additional Editor Comments (optional):

Reviewers' comments:

Reviewer's Responses to Questions

**Comments to the Author**

1. If the authors have adequately addressed your comments raised in a previous round of review and you feel that this manuscript is now acceptable for publication, you may indicate that here to bypass the “Comments to the Author” section, enter your conflict of interest statement in the “Confidential to Editor” section, and submit your "Accept" recommendation.

Reviewer #1: All comments have been addressed

Reviewer #2: All comments have been addressed

2. Is the manuscript technically sound, and do the data support the conclusions?

Reviewer #1: Yes

Reviewer #2: Yes

3. Has the statistical analysis been performed appropriately and rigorously? 

Reviewer #1: Yes

Reviewer #2: Yes

4. Have the authors made all data underlying the findings in their manuscript fully available?

Reviewer #1: Yes

Reviewer #2: Yes

5. Is the manuscript presented in an intelligible fashion and written in standard English?

Reviewer #1: Yes

Reviewer #2: Yes

6. Review Comments to the Author

Reviewer #1: (No Response)

Reviewer #2: The revised manuscript is well-presented and all comments have been addressed. This manuscript is now acceptable for publication.

7. PLOS authors have the option to publish the peer review history of their article (what does this mean?). If published, this will include your full peer review and any attached files.

Reviewer #1: No

Reviewer #2: No

---

## [Editor Report · Acceptance letter]

11 Sep 2020

PONE-D-20-05326R1 

Non-Steroidal Anti-Inflammatory Drugs (NSAIDs): Usage and Co-Prescription with Other Potentially Interacting Drugs in Elderly: a cross-sectional study 

Dear Dr. Abdu:

I'm pleased to inform you that your manuscript has been deemed suitable for publication in PLOS ONE. Congratulations! Your manuscript is now with our production department. 

Kind regards, 

on behalf of

Prof. Jinn-Moon Yang 

Academic Editor

PLOS ONE